# Decomposition of Glucose-Sensitive Layer-by-Layer Films Using Hemin, DNA, and Glucose Oxidase

**DOI:** 10.3390/polym12020319

**Published:** 2020-02-04

**Authors:** Kentaro Yoshida, Yu Kashimura, Toshio Kamijo, Tetsuya Ono, Takenori Dairaku, Takaya Sato, Yoshitomo Kashiwagi, Katsuhiko Sato

**Affiliations:** 1School of Pharmaceutical Sciences, Ohu University 31-1 Misumido, Tomita-machi, Koriyama, Fukushima 963-8611, Japan; 2Department of Creative Engineering, National Institute of Technology, Tsuruoka College, 104 Sawada, Inooka, Tsuruoka 997-8511, Japan; 3Graduate School of Pharmaceutical Sciences, Tohoku University, 6-3 Aoba, Aramaki, Aoba-ku, Sendai 980-8578, Japan

**Keywords:** hydrogen peroxide response, layer-by-layer, multilayer thin film, glucose sensitive, stimuli-sensitive

## Abstract

Glucose-sensitive films were prepared through the layer-by-layer (LbL) deposition of hemin-modified poly(ethyleneimine) (H-PEI) solution and DNA solution (containing glucose oxidase (GOx)). H-PEI/DNA + GOx multilayer films were constructed using electrostatic interactions. The (H-PEI/DNA + GOx)_5_ film was then partially decomposed by hydrogen peroxide (H_2_O_2_). The mechanism for the decomposition of the LbL film was considered to involve more reactive oxygen species (ROS) that were formed by the reaction of hemin and H_2_O_2_, which then caused nonspecific DNA cleavage. In addition, GOx present in the LbL films reacts with glucose to generate hydrogen peroxide. Therefore, decomposition of the (H-PEI/DNA + GOx)_5_ film was observed when the thin film was immersed in a glucose solution. (H-PEI/DNA + GOx)_5_ films exposed to a glucose solution for periods of 24, 48 72, and 96 h indicated that the decomposition of the film increased with the time to 9.97%, 16.3%, 23.1%, and 30.5%, respectively. The rate of LbL film decomposition increased with the glucose concentration. At pH and ionic strengths close to physiological conditions, it was possible to slowly decompose the LbL film at low glucose concentrations of 1–10 mM.

## 1. Introduction

Layer-by-layer (LbL) films have previously been prepared by the alternate deposition of polyelectrolytes (polycation and polyanion) on a solid surface, assisted by electrostatic interaction [1,2,3]. Various other interactions have also been recently employed to construct LbL films, such as hydrogen bonding [4,5] and sugar-lectin binding [6,7]. The materials employed for this purpose have included synthetic polymers [8,9], polysaccharides [10,11,12], protein [13,14,15], and DNA [16,17]. Such layered multilayer films have found application in separation and purification [18,19], sensors [20,21], and drug delivery systems (DDSs) [22,23,24].

We have recently reported that hydrogen peroxide (H_2_O_2_) induced the decomposition of LbL films composed of hemin-modified poly(ethyleneimine) (H-PEI) and DNA [25]. Hemin is an iron porphyrin molecule and the iron porphyrin produces more reactive oxygen species (ROS), such as hydroxy radicals (OH), by reaction with H_2_O_2_ [26,27]. The ROS cause non-specific DNA cleavage [28,29,30], and H-PEI/DNA LbL film was consequently decomposed by the addition of H_2_O_2_. H_2_O_2_ is generated by the reaction of substrates and oxidases. LbL films composed of oxidases have been applied in biosensors and stimuli-responsive devices [31,32,33]. In the present work, we report the preparation of thin LbL films consisting of H-PEI and DNA with glucose oxidase (GOx), and the glucose-induced decomposition of these LbL film (Figure 1). Glucose-sensitive materials respond to the blood glucose level of a diabetic patient; therefore, LbL films composed of GOx can be applied to an insulin DDS [34,35]. The decomposition of LbL films reported in this study is slow due to the stepwise reactions (enzymatic reaction, generation of ROS by hemin, and nonspecific cleavage of DNA). If decomposition of the membrane is slow, then the drug encapsulated in a capsule membrane can be released gradually. The glucose-induced decomposition of the LbL film in this work was achieved by the addition of glucose and the decomposition’s dependence on the concentration of glucose was investigated.

## 2. Materials and Methods

### 2.1. Materials

Hemin and poly(ethyleneimine) (PEI, molecular weight; 60,000–80,000) were obtained from Tokyo Chemical Industry Co. (Tokyo, Japan). PEI has a random branched structure with the ratio of primary, secondary, and tertiary amino groups being nominally ca. 1:2:1. DNA (calf thymus) was purchased from Nacalai Tesque Inc. (Tokyo, Japan). GOx was obtained from Sigma-Aldrich Chemical Co. (St. Louis, MO, USA). All other reagents used were of the highest grade and used without further purification.

H-PEI was synthesized as follows. PEI (100 mg) and hemin (37.9 mg) were dissolved in dimethyl sulfoxide, to which N-hydroxysuccinimide (8.02 mg) and 1-ethyl-3-(3-dimethylaminopropyl) carbodiimide hydrochloride (13.4 mg) were added at 4 °C. After 24 h, the reaction mixture was purified by dialyzing with water for 3 days (dialysis tubing, nominal MWCO: 3500, Hampton, NH, USA) and then freeze-dried. The content of hemin residues was 2.2–2.6% (molar ratio of hemin to amine), as determined from hemin-derived absorbance (at 390 nm) using UV-vis absorption spectroscopy. Figure 2 shows the chemical structures of hemin, PEI, and H-PEI.

### 2.2. Apparatus

A quartz crystal microbalance (QCM; eQCM 10M Garmry, Warminster, UK) was employed for the gravimetric analysis of LbL films consisting of H-PEI and DNA. An 8 MHz AT-cut quartz resonator coated with a gold (Au) layer (surface area 0.2 cm^2^) was used as a probe. The Au surface layer of the quartz resonator was cleaned using piranha solution (a mixture of H_2_O_2_ and H_2_SO_4_, 1:3 v/v) and thoroughly rinsed in pure water before use (CAUTION: piranha solution should be handled with extreme care). All QCM operations used flow QCM cells (cell volume ca. 120 µL; EQCM flow cell kit, BAS, Tokyo, Japan). Atomic force microscopy (AFM; AFM5200S, Hitachi High-Technologies Co., Tokyo, Japan) images were acquired in contact mode at room temperature in air. UV-vis spectroscopy measurements were conducted using a V-560 (Jasco, Tokyo, Japan) spectrometer.

### 2.3. Preparation of LbL Films

H-PEI/DNA films were prepared on the cleaned quartz resonator for QCM analysis. The quartz resonator was immersed in 0.1 mg/mL H-PEI solution in 10 mM HEPES buffer containing 150 mM NaCl (pH 7.4) for 15 min to deposit the first H-PEI layer by the hydrophobic force of attraction. After being rinsed in buffer for 5 min to remove any weakly adsorbed H-PEI, the quartz resonator was immersed in 0.1 mg/mL DNA solution for 15 min to deposit DNA by electrostatic interaction. At that time, 0.1 mg/mL GOx was mixed in the DNA solution. A second H-PEI layer was deposited similarly on the surface of the quartz resonator. The deposition steps were repeated to build up LbL films. Circular glass slides (18 mm diameter) and quartz slides (50 × 9 × 1 mm) with LbL films were prepared in the same manner. UV-vis absorption spectra of the LbL films in the working buffer were recorded on a UV-vis spectrometer. For dry AFM observations, the circular glass slides used to prepare each of the (H-PEI/DNA + GOx)_5_ films were rinsed with milli-Q water and dried for 24 h in a desiccator. AFM images were acquired in AC mode using an Arrow-NCR probe (Toyo Corporation, Tokyo, Japan) at room temperature in air.

### 2.4. Decomposition of LbL Films

The H_2_O_2_-induced decomposition of the (H-PEI/DNA + GOx)_5_ films was studied using UV-vis absorption spectroscopy. The LbL films prepared on quartz slides were exposed to 1, 10, and 100 mM H_2_O_2_ solution (pH 7.4) for 30, 60, 90, 120, 180, and 240 min, and then rinsed with the working buffer for 5 min. The films were measured for absorption at a particular time and then subsequently immersed in the H_2_O_2_ solution for the next exposure step.

The glucose-induced decomposition of the (H-PEI/DNA + GOx)_5_ films was monitored in the same manner. The LbL films prepared on quartz slides were exposed to 1, 10, and 100 mM glucose solution (pH 7.4) for 0.5, 1, 1.5, 2, 3, 4, 5, 6, 8, 10, 24, 30, 35, and 48 h. The glucose-induced decomposition of the LbL films was studied by monitoring the resonance frequency change (Δ*F*) of the 5-bilayer film-coated quartz resonator in the flow-through cell of the QCM. After rinsing the film-deposited probe with buffer for 5 min, the steady-state frequency was recorded. The film-deposited probe was measured for frequency at a particular time and then subsequently immersed in the glucose solution for the next exposure step.

## 3. Results and Discussion

Figure 3 shows the change in the resonance frequency (Δ*F*) of the QCM when the quartz resonator was immersed in the H-PEI solution and DNA/GOx mixed solution. The Δ*F* values decreased with the deposition of both H-PEI and the DNA/GOx mixture, which indicated that the (H-PEI/DNA + GOx) film was successfully formed on the surface of the quartz resonator. From the flow QCM data, the change in the resonance frequency of the (H-PEI/DNA + GOx)_5_ film was −1970 ± 342 Hz (n = 3). It is considered that positively charged H-PEI and negatively charged DNA and GOx are deposited by electrostatic attraction, which builds up the LbL film on the quartz resonator surface. There have been other reports on the preparation of LbL films by electrostatic interaction using QCM [36]. LbL films using DNA or GOx driven by electrostatic interactions have been reported by Zhang et al. [37,38] and Kakade et al. It is considered that DNA and GOx are both present in the negatively charged layers because DNA and GOx adsorb to positively charged polymers.

Figure 4 shows an AFM image and a depth profile of a dried (H-PEI/DNA + GOx)_5_ film. The circular glass slides (18 mm diameter) used to prepare each of the (H-PEI/DNA + GOx)_5_ films were rinsed with milli-Q water and dried for 24 h in a desiccator. The thicknesses of the LbL films were determined by scratching the film-coated glass slide using a cutter and performing AFM depth profile scans over the scratch. The thicknesses of the (H-PEI/DNA + GOx)_5_ films were estimated to be 45.68 ± 15.95 nm.

Figure 5 show UV-vis absorption spectra for the (H-PEI/DNA + GOx)_5_ film and exposure to 100 mM H_2_O_2_ solution in 10 mM HEPES buffer containing 150 mM NaCl (pH 7.4), respectively. The hemin has an absorption at 390 nm (Appendix A). H-PEI is adsorbed on the surface of the DNA layers, and the LbL film has an absorption maximum derived from hemin. The disappearance of the absorption at 390 nm was confirmed after the (H-PEI/DNA + GOx)_5_ film was exposed to 100 mM H_2_O_2_ solution for 1 h. The iron porphyrin is known to be oxidized by H_2_O_2_, according to Equation (1) [39,40].
Porphyrin-Fe(III) + H_2_O_2_ → Porphyrin-Fe(IV) = O + OH + H^+^(1)

The oxidation of iron porphyrin significantly reduces the absorption at 390 nm (Appendix A). The iron porphyrin contained in hemin produces ROS. The H_2_O_2_ causes the hemin to be degraded, so that the absorption maximum derived from hemin in the film was significantly reduced. Therefore, the absorption at 390 nm is not suitable for the evaluation of LbL film decomposition. The absorbance of the (H-PEI/DNA + GOx)_5_ film at 260 nm significantly decreased when immersed in 100 mM H_2_O_2_ solution. The DNA solution has an absorption at 260 nm, and there is no significant change in absorption at 260 nm after hydrogen peroxide treatment (Appendix A). The absorbance coefficient of GOx is small compared to DNA and hemin (Appendix A). Therefore, GOx has a negligible effect on the absorbance, unless the concentration is high. The oxidation of the iron porphyrin slightly reduces the absorbance contribution at 260 nm, whereas the contribution of the (H-PEI/DNA + GOx)_5_ film to the absorbance is more significant (Appendix A). If ROS are generated near the LbL films, then nonspecific cleavage of DNA is triggered. Yoshida et al. reported the decomposition of H-PEI/DNA nanofilm by H_2_O_2_ [25]. Similarly, the decomposition of the (H-PEI/DNA + GOx)_5_ films by H_2_O_2_ resulted in a decrease in the absorption maximum derived from DNA. Therefore, changes in the absorption at 260 nm can be an indicator of the degradability of the (H-PEI/DNA + GOx)_5_ film.

The H_2_O_2_-induced decomposition of (H-PEI/DNA + GOx)_5_ films was investigated using UV-vis absorption spectroscopy (Figure 6). The absorbance at 260 nm is attributed to hemin, DNA, and GOx. Therefore, if the film is decomposed, the constituent components of the film on the quartz slides are lost, and the absorbance at 260 nm decreases. Therefore, the decomposition of the LbL film was evaluated using an absorbance ratio. The extent of the H_2_O_2_-induced decomposition of the LbL films was determined using Equation (2).
(2)$$\mathrm{Ab}s_{t}/\mathrm{Ab}s_{0}\times 100=\frac{\begin{array}{c} \mathrm{absorbance}\;\mathrm{at}\;260\;\mathrm{nm}\;\mathrm{of}\;(H-\;\mathrm{PEI}/\mathrm{DNA}\;+\mathrm{GOx}{)}_{5}\;\mathrm{film}\\ \mathrm{when}\;\mathrm{immersed}\;\mathrm{in}\;\mathrm{hydorogen}\;\mathrm{peroxide}\;\mathrm{solution}\;\mathrm{for}\;t\;\min \end{array}}{\mathrm{absorbance}\;\mathrm{at}\;260\;\mathrm{nm}\;\mathrm{of}\;(H-\;\mathrm{PEI}/\mathrm{DNA}\;+\mathrm{GOx}{)}_{5}\;\mathrm{film}}\times 100$$

When the (H-PEI/DNA + GOx)_5_ film was exposed to the buffer for a long time, the absorbance did not change. On the other hand, the (H-PEI/DNA + GOx)_5_ film immersed in H_2_O_2_ solution showed a decrease in the absorbance ratio, and the decrease was larger as the H_2_O_2_ concentration was higher. Significant decrease in the absorbance ratio can thus indicate the decomposition of the LbL films by H_2_O_2_. In addition, the decomposition of the membrane is affected by the H_2_O_2_ concentration. If the same method is used, then the glucose-induced decomposition of (H-PEI/DNA + GOx)_5_ film can be evaluated.

The glucose-induced decomposition of (H-PEI/DNA + GOx)_5_ films was investigated using UV-vis spectroscopy (Figure 7). The glucose-induced LbL films were determined using Equation (3).
(3)$$\mathrm{Ab}s_{t}/\mathrm{Ab}s_{0}\times 100=\frac{\begin{array}{c} \mathrm{absorbance}\;\mathrm{at}\;260\;\mathrm{nm}\;\mathrm{of}\;(H-\;\mathrm{PEI}/\mathrm{DNA}\;+\mathrm{GOx}{)}_{5}\;\mathrm{film}\\ \mathrm{when}\;\mathrm{immersed}\;\mathrm{in}\;\mathrm{glucose}\;\mathrm{solution}\;\mathrm{for}\;t\;\min \end{array}}{\mathrm{absorbance}\;\mathrm{at}\;260\;\mathrm{nm}\;\mathrm{of}\;(H-\;\mathrm{PEI}/\mathrm{DNA}\;+\mathrm{GOx}{)}_{5}\;\mathrm{film}}\times 100$$

When (H-PEI/DNA + GOx)_5_ films were exposed to 1, 10, and 100 mM glucose solutions for 48 h, the absorbance ratio was 76.1%, 70.2%, and 64.0%, respectively. Hemin is an iron porphyrin molecule and is an active cofactor for various enzymes, such as catalase and peroxidase [28]. GOx present in the LbL films reacts with glucose to generate gluconic acid and H_2_O_2_. H_2_O_2_ present in the LbL films reacts with hemin to generate ROS. The nonspecific cleavage of DNA by ROS promotes the partial degradation of the LbL films. In addition, the rate of LbL film decomposition increased with the glucose concentration. The decomposition rate of the film immersed in the glucose solution was slower than that immersed in the H_2_O_2_ solution. However, the influence of the glucose concentration on the decomposition of the (H-PEI/DNA + GOx)_5_ film was less than that from the H_2_O_2_ concentration. GOx and glucose are enzymatic reactions; therefore, H_2_O_2_ is generated slowly and the degradation of the LbL film is delayed. Therefore, at pH and ionic strengths close to physiological conditions, it is possible to slowly decompose the LbL films at a low glucose concentration (1–10 mM).

The glucose-induced decomposition of (H-PEI/DNA + GOx)_5_ films was investigated using the QCM (Figure 8). The extent of the decomposition was calculated from the change in the resonance frequency (Equation (4)).
(4)$$\mathrm{Decomposed}\;\left(\%\right)=\left(1-\frac{\Delta F_{t}}{\Delta F}\right)\times 100$$

Δ*F*: frequency change of the (H − PEI/DNA)_5_ films

Δ*F_t_*: frequency change difference between blank and (H − PEI/DNA)_5_ films immersed in glucose solution for *t* time

When the (H-PEI/DNA + GOx)_5_ film was exposed to 100 mM glucose solution for 24, 48, 72, and 96 h, the extent of decomposition was 9.97%, 16.3%, 23.1%, and 30.5%, respectively. However, the results of Figure 8 show less decomposition of LbL films than expected compared to Figure 7. Compared to QCM operation, the amount of decomposition of LbL films immersed in glucose solution during UV operation is greater. The absorbance ratio is influenced by hemin oxidation; therefore, there was a difference in the amount of membrane degradation. The glucose-induced decomposition of LbL films reported by Sato et al. occurred for 1 h when glucose was added [41]. On the other hand, from the QCM results, a longer immersion time in 100 mM glucose solution resulted in greater decomposition of the LbL films. Depending on the glucose concentration, it is possible to slowly decompose the (H-PEI/DNA + GOx)_5_ film.

## 4. Conclusions

(H-PEI/DNA + GOx)_5_ LbL films were prepared by alternate immersion of a substrate in H-PEI solution and DNA solution (containing GOx). When the (H-PEI/DNA + GOx)_5_ films were immersed in a H_2_O_2_ solution, partial decomposition of the LbL films was observed. The iron porphyrin in hemin produces more ROS from the reaction with H_2_O_2_ [26,27]. The ROS cause non-specific DNA cleavage; therefore, the decomposition of the LbL films composed of DNA was observed. Furthermore, partial degradation was observed when the membrane was immersed in a glucose solution. GOx present in LbL films produced H_2_O_2_ from glucose. These LbL films decomposed under physiological conditions with various glucose concentrations, which suggests that a glucose stimuli-responsive nanofilm could be realized. We have developed other glucose- and pH-responsive thin films [42,43]; however, the (H-PEI/DNA + GOx)_5_ film has a very slow decomposition. If a drug could be encapsulated in this thin film, then there is a possibility that a system capable of drug release over a long time period, depending on the substrate, could be realized.

## Figures and Tables

**Figure 1 polymers-12-00319-f001:**
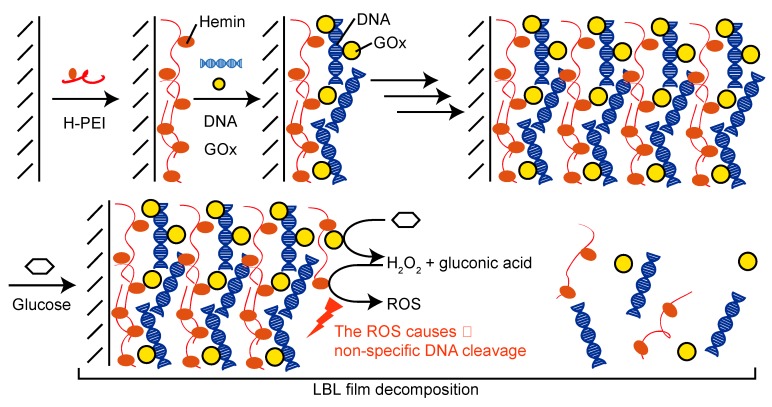
Preparation and decomposition of glucose-sensitive layer-by-layer (LbL) films composed of hemin, DNA, and glucose oxidase (GOx).

**Figure 2 polymers-12-00319-f002:**
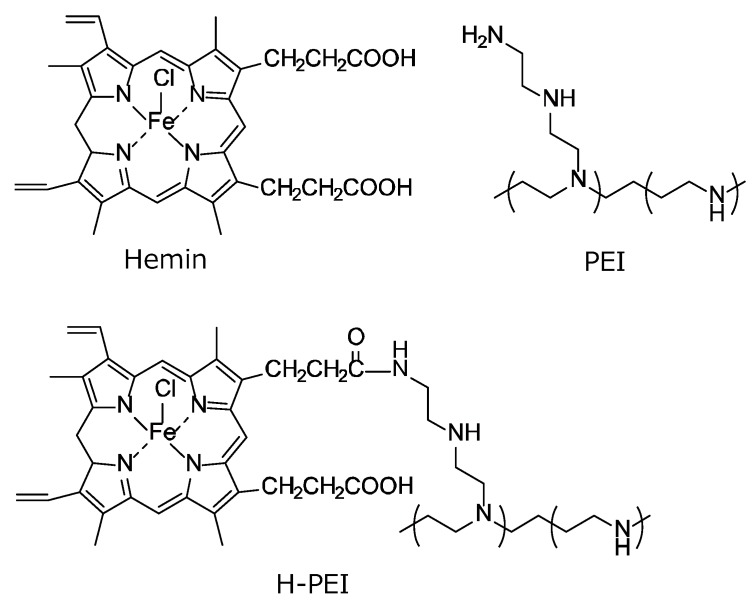
Chemical structures of hemin, poly(ethyleneimine) (PEI), and hemin-modified poly(ethyleneimine) (H-PEI).

**Figure 3 polymers-12-00319-f003:**
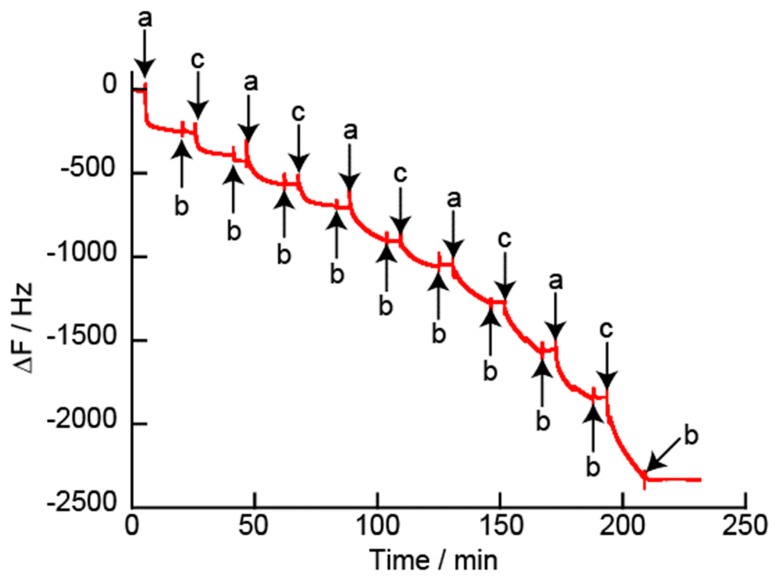
Change in the QCM frequency for the deposition of H-PEI/DNA + GOx)_5_ film at pH 7.4. The resonator was exposed to (**a**) 0.1 mg/mL H-PEI, (**b**) 10 mM HEPES buffer solution, and (**c**) 0.1 mg/mL DNA (containing 0.1 mg/mL GOx).

**Figure 4 polymers-12-00319-f004:**
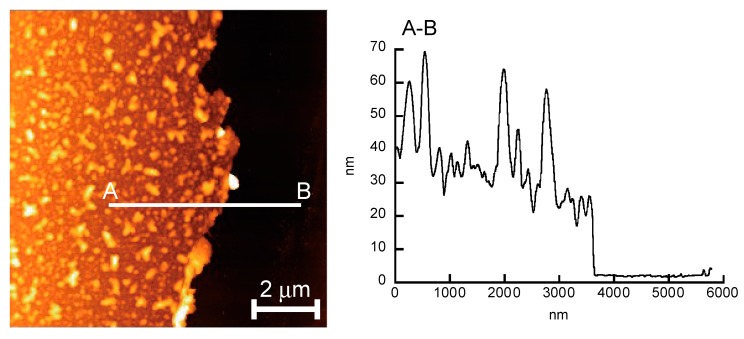
AFM image of the dried (H-PEI/DNA + GOx)_5_ film.

**Figure 5 polymers-12-00319-f005:**
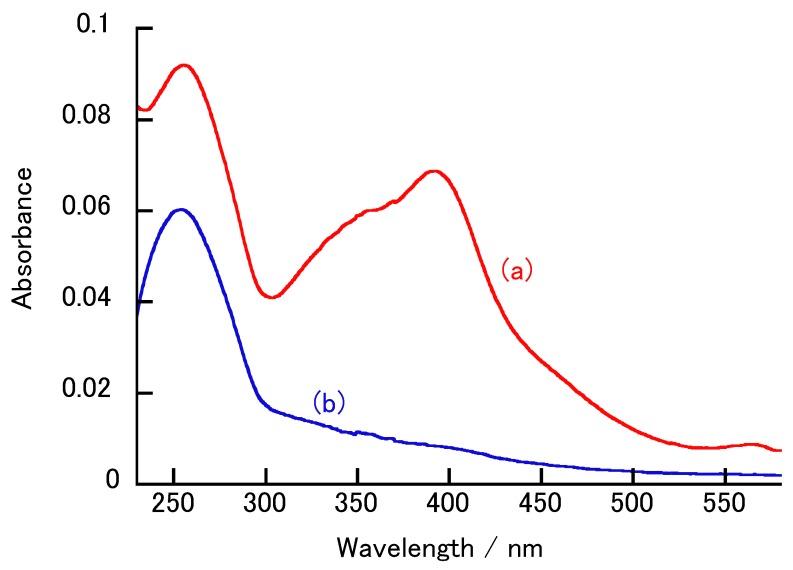
UV-vis absorption spectra for the (H-PEI/DNA + GOx)_5_ film (**a**) before and (**b**) after exposure to 100 mM H_2_O_2_ solution (pH 7.4) for 1 h.

**Figure 6 polymers-12-00319-f006:**
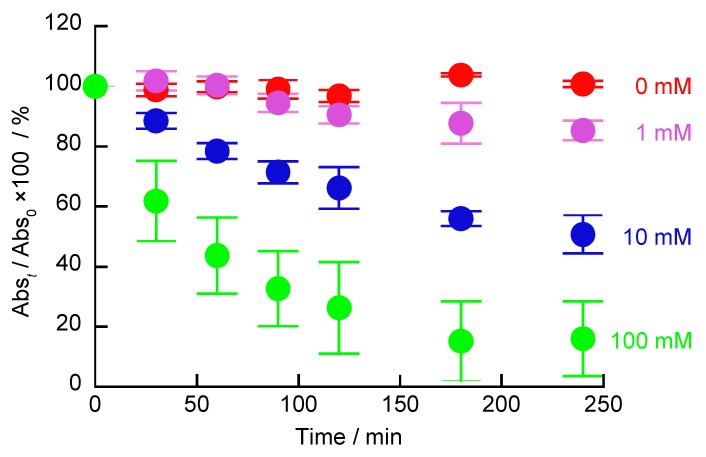
H_2_O_2_-induced decomposition of (H-PEI/DNA + GOx)_5_ films investigated using UV-vis absorption spectroscopy. The LbL films were exposed to 0 (red), 1 (purple), 10 (blue), and 100 mM (green) H_2_O_2_ solutions (pH 7.4) for up to 240 min. Error bars represent standard deviation (n = 4).

**Figure 7 polymers-12-00319-f007:**
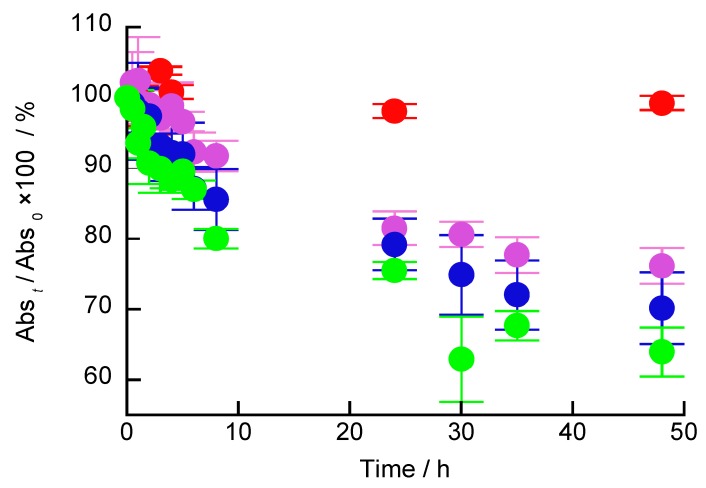
Glucose-induced decomposition of (H-PEI/DNA + GOx)_5_ films investigated using UV-vis absorption spectroscopy. The LbL films were exposed to 0 mM (red), 1 mM (purple), 10 mM (blue), and 100 mM (green) glucose solutions (pH 7.4) for up to 48 h. Error bars represent standard deviation (0 mM; n = 4, 1, 10, and 100 mM; n = 3,).

**Figure 8 polymers-12-00319-f008:**
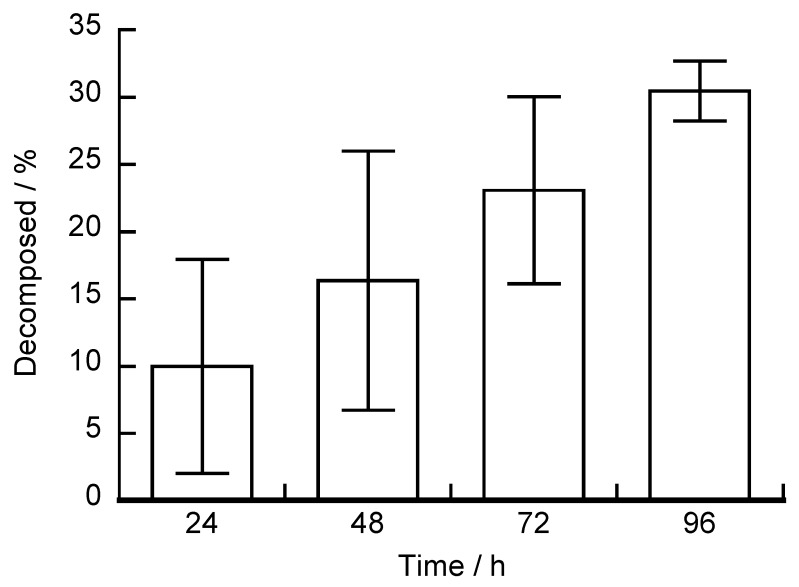
Glucose-induced decomposition of (H-PEI/DNA + GOx)_5_ films was investigated using a QCM. The LbL film was exposed to 100 mM glucose solution (pH 7.4) for up to 96 h. Error bars represent standard deviation (n = 3).

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
