# Peer review of "Decomposition of Glucose-Sensitive Layer-by-Layer Films Using Hemin, DNA, and Glucose Oxidase"

_polymers, 2020, doi:10.3390/polym12020319_

Round 1

Reviewer 1 Report

In this manuscript, the authors introduce the production of glucose-sensitive films composed of hemin modified poly(ethyleneimine) (H-PEI), DNA and glucose oxidase(GOx). In the glucose environment, GOx oxidizes glucose to generate H2O2, and the generated H2O2 reacts with hemin to generate ROS, which causes DNA to cleavage, resulting in partial decomposition of films, which might be interesting to many readers.

However, it still remains some concerns.

(1) I am curious to check the potential application of this glucose sensitive films? If for glucose detection, there are currently many reports on that. It might be worthy to highlight more on the advanced design in this report. And if for in vivo detection of glucose level, would it be better to add some application data at the cellular or animal level in this article?

(2) There are many glucose-sensitive systems, the authors can explain why DNA and PEI are used to form the films, and what role does DNA play in this system, just as film stents (why not other materials (for example, disulfide) with low cost)? If it is used as film stents, is it safe to use a large amount of Exogenous DNA in vivo?

(3) The authors mentioned that in the presence of glucose, and a series of reactions occur in the films to produce ROS, which causes the DNA strand to break and films degradation. I am curious to check whether the authors quantitatively measure the generation of ROS.

(4) It is better to provide data on the characterization of the film, such as the appearance or SEM images of the films.

(5) Language issues. For example, Abstract section, line 18, “The mechanism for the decomposition of the LbL film was considered to involve a more reactive oxygen species (ROS) that was formed by the reaction of hemin and H2O2, which then caused nonspecific DNA cleavage. GOx present in the LbL films reacts with glucose to generate hydrogen peroxide.” The logical order of this sentence might be problematic.

Author Response

Thank you for your kind suggestion of revision of our manuscript (polymers- 702586).

We have revised the manuscript according to reviewer’s comments. All revisions made are marked in red in the revised manuscript. Our responses are as follows.

To Review 1

(1) I am curious to check the potential application of this glucose sensitive films? If for glucose detection, there are currently many reports on that. It might be worthy to highlight more on the advanced design in this report. And if for in vivo detection of glucose level, would it be better to add some application data at the cellular or animal level in this article?

[Response] We are studying thin films that decompose in response to stimuli. In the stimulus response, we focused on thin films that decompose slowly in addition to those that decompose quickly. We thought that using a stepwise reaction would slow down the decomposition of LbL films (Line 49-54). In vivo level research is needed in the future, but I would like to summarize the results of this study before that.

(2) There are many glucose-sensitive systems, the authors can explain why DNA and PEI are used to form the films, and what role does DNA play in this system, just as film stents (why not other materials (for example, disulfide) with low cost)? If it is used as film stents, is it safe to use a large amount of Exogenous DNA in vivo?

[Response] Previous studies have shown that (H-PEI/DNA) film are easily decomposed by hydrogen peroxide [25]. The ROS causes non-specific DNA cleavage, DNA (calf thymus) was used as a material for the LbL films because of its low cost. The use of large amounts of exogenous DNA is high risk, but using simple nucleotide chains may reduce the risk. Cell-level studies of injecting simple nucleotide chains have also been reported (J. AM. CHEM. SOC. 2009, 131, 15761–15768).

(3) The authors mentioned that in the presence of glucose, and a series of reactions occur in the films to produce ROS, which causes the DNA strand to break and films degradation. I am curious to check whether the authors quantitatively measure the generation of ROS.

[Response] Since it is difficult to track the ROS generated in the thin film every hour, we do not measure ROS quantitatively.

(4) It is better to provide data on the characterization of the film, such as the appearance or SEM images of the films.

[Response] We agree the suggestion. But,the environment for measuring SEM is not ready. However, we are considering encapsulation of this film, so we want to measure it at that time.

(5) Language issues. For example, Abstract section, line 18, “The mechanism for the decomposition of the LbL film was considered to involve a more reactive oxygen species (ROS) that was formed by the reaction of hemin and H2O2, which then caused nonspecific DNA cleavage. GOx present in the LbL films reacts with glucose to generate hydrogen peroxide.” The logical order of this sentence might be problematic.

[Response] We agree. The text has been corrected (lines 21). The text was added so that the mechanism of decomposition by hydrogen peroxide and the decomposition by immersion in a glucose solution can be considered separately.

Reviewer 2 Report

The work by Yoshida et al. presents a study of the fabrication of LbL multilayers with potential application as biosensors. The work is well performed and the results are sound. I have to concernings

-Authors should include among their references: Soft Matter, 2009, 5, 2130-2142. This is probably one of the most relevant works for understanding the LbL assembly.

-Authors cannot use a QCM measuring only the fundamental frequency for quantification, first they do not have information on the mechanical properties of the films, thus they cannot be sure that the Sauerbrey approach works, and then the fundamental frequency is affected from the piezoelectric stiffness, thus the results are not alway reproducible. Thus, the QCM results should be presented in a qualitative way without any reference to the deposited mass.

Author Response

Thank you for your kind suggestion of revision of our manuscript (polymers- 702586).

We have revised the manuscript according to reviewer’s comments. All revisions made are marked in red in the revised manuscript. Our responses are as follows.

To Review 2

(1)Authors should include among their references: Soft Matter, 2009, 5, 2130-2142. This is probably one of the most relevant works for understanding the LbL assembly.

[Response] We agree the suggestion. It was very helpful. Add to reference..

(2)Authors cannot use a QCM measuring only the fundamental frequency for quantification, first they do not have information on the mechanical properties of the films, thus they cannot be sure that the Sauerbrey approach works, and then the fundamental frequency is affected from the piezoelectric stiffness, thus the results are not alway reproducible. Thus, the QCM results should be presented in a qualitative way without any reference to the deposited mass.

[Response] We agree the suggestion. Except for the description of the weight of the membrane, the description has been changed to the description of the frequency change.

Round 2

Reviewer 1 Report

The authors have improved the manuscript and solved my previous concerns. Hence, I would like to recommend the publication of this version.